# Sauron U-Net: Simple automated redundancy elimination in medical image segmentation via filter pruning

## Abstract

We present Sauron, a filter pruning method that eliminates redundant feature maps by discarding the corresponding filters with automatically-adjusted layer-specific thresholds. Furthermore, Sauron minimizes a regularization term that, as we show with various metrics, promotes the formation of feature maps clusters. In contrast to most filter pruning methods, Sauron is single-phase, similarly to typical neural network optimization, requiring fewer hyperparameters and design decisions. Additionally, unlike other cluster-based approaches, our method does not require pre-selecting the number of clusters, which is non-trivial to determine and varies across layers. We evaluated Sauron and three state-of-the-art filter pruning methods on three medical image segmentation tasks. This is an area where filter pruning has received little attention and where it can help building efficient models for medical grade computers that cannot use cloud services due to privacy considerations. Sauron achieved models with higher performance and pruning rate than the competing pruning methods. Additionally, since Sauron removes filters during training, its optimization accelerated over time. Finally, we show that the feature maps of a Sauron-pruned model were highly interpretable. The Sauron code is publicly available at `https://github.com/blindedrepository`.

## 1 Introduction

Pruning is the process of eliminating unnecessary parameters to obtain compact models and accelerate their inference. There are two main strategies for pruning convolutional neural networks (CNNs): weight pruning and filter pruning. In weight pruning, weights for unimportant connections are zeroed without consideration of the network structure, leading, in practice, to sparse weight matrices [21, 12, 10, 11, 40]. On the other hand, filter pruning methods eliminate CNNs filters directly. Thus, unlike weight-pruned models, utilizing filter-pruned networks efficiently requires no specialized hardware or software [7, 35]. Most pruning methods have been developed or evaluated exclusively for natural image classification. Other tasks, such as medical image segmentation, have received significantly less attention [32]. In medical imaging, small models can enable computationally-limited medical grade computers to segment medical images that cannot be uploaded to a cloud server due to privacy reasons. Moreover, models with a few filters can be easier to interpret than large models, which is crucial not only in clinical applications but also in research. Motivated by these possibilities, we propose a filter pruning method called Sauron that generates small CNNs. We demonstrate its

application to prune U-Net-like networks [36], bringing together filter pruning and medical image segmentation.

Sauron applies filter pruning during optimization in a *single phase*, while most filter pruning frameworks consist of three distinct phases: Pre-training the model, pruning its filters, and fine-tuning to compensate for the loss of accuracy (or re-training from scratch [27, 6]). Other approaches combine pruning with training [46, 48, 13, 38] or fine-tuning [29, 26], resulting in two-phase frameworks, and other methods repeat these phases multiple times [46, 29, 4]. Single-phase filter pruning methods [48], such as Sauron, are advantageous since they require fewer hyperparameters and design decisions, including the number of epochs for training and fine-tuning, pruning iterations, or whether to combine pruning with training or fine-tuning. In particular, Sauron does not insert additional parameters into the optimized architecture to identify filter candidates for pruning, such as channel importance masks [6, 29, 15, 26, 16]. This avoids potential optimization hindrance and requires less extra training time and GPU memory.

Sauron facilitates and promotes the formation of feature map clusters by optimizing a regularization term, and, unlike previous cluster-based approaches [9, 13, 4], Sauron does not enforce the number of these clusters. Since these clusters vary depending on the training data and across layers, the optimal number of feature maps per cluster is likely to differ. Thus, determining the number of clusters is not trivial and may limit the accuracy and the pruning rate.

Our specific contributions are the following:

- We introduce Sauron, a single-phase filter pruning method that resembles the typical CNN optimization, making it easier to use, and that does not add any additional parameters to the optimized architecture.

- We show that Sauron promotes the formation of feature map clusters by optimizing a regularization term.

- We compare Sauron to other methods on three medical image segmentation tasks, where Sauron resulted in more accurate and compressed models.

- We show that the feature maps generated by a model pruned with Sauron were highly interpretable.

- We publish Sauron and the code to run all our experiments at `https://github.com/blindedrepository`.

## 2 Previous work

**Filter importance** Most filter pruning approaches rely on ranking filters to eliminate the unimportant filters. The number of eliminated filters can be determined by either a fixed [3] or an adaptive threshold [38]. Filter importance can be found via particle filtering [3] or it can be computed via heuristic relying on measures such as $L_p$ norms [23, 44, 38], entropy [28], or post-pruning accuracy [1]. Pruning methods can include extra terms in the loss function, such as group sparsity constraints, although these extra terms guarantee no sparsity in CNNs [45]. Other methods aim to learn filter importance by incorporating channel importance masks into CNNs' architectures [6, 29, 15, 26, 16]. However, these adjustments modify the architectures to be optimized, increasing the required GPU memory during training, optimization time, and potentially hindering the optimization. Alternatively, other methods consider the scaling factor of batch normalization layers as channel importance [45, 48], but in e.g. medical image segmentation, batch normalization is occasionally replaced by other normalization layers due to the small mini-batch size [18].

**Difference minimization** Methods that remove filters while trying to preserve characteristics such as classification accuracy [27], Taylor-expansion-approximated loss [46], and the feature maps [47, 42, 44, 30] of the original unpruned models. A disadvantage of these methods is that they require a large GPU memory to avoid loading and unloading the models in memory constantly, which would

---

**Algorithm 1** Sauron

---

**Input:** training data: $\mathcal{D}$.

1: **Given**: $\lambda$, maximum threshold $\tau_{max}$, $epochs$, percentage of pruned filters $\mu$, patience $\rho$, number of steps $\kappa$.
2: **Initialize**: model's weights $\mathbf{W} \leftarrow \{\mathbf{W}^l, 1 \le l \le L\}$, layer-specific thresholds $\boldsymbol{\tau} \leftarrow \{\tau_l = 0, 1 \le l \le L\}$
3: **for** $e = 1; e \le epochs$ **do**
4:     **for** $b = 1; b \le N$ **do** *# Mini batches*
5:        Compute predictions $\hat{\boldsymbol{y}}$, loss $\mathcal{L}$, $\delta_{opt}$ (Eq. (2)), $\boldsymbol{\delta}_{prune}$ (Eq. (3)) *# Forward pass*
6:        Update $\boldsymbol{\theta}$                                       *# Backward pass*
7:     **end for**
8:     **for** $l = 1; l \le L$ **do**                                  *# Pruning step*
9:        *## Procedure 1: Increasing $\tau_l$ ##*
10:        C1: Training loss is converging; C2: Validation loss is not improving
11:        C3: Less than $\mu\%$ of filters pruned in $(e - 1)$; C4: $\tau_l$ has not increased in last $\rho$ epochs
12:        **if** (C1 $\wedge$ C2 $\wedge$ C3 $\wedge$ C4) $\wedge$ ($\tau_l < \tau_{max}$) **then**
13:           $\tau_l \leftarrow \tau_l + \tau_{max}/\kappa$
14:        **end if**
15:        *## Procedure 2: Pruning ##*
16:        $\mathbf{W}^l \leftarrow \{\mathbf{W}^l : \boldsymbol{d}^l > \tau_l\}$
17:     **end for**
18: **end for**

**Output:** Pruned CNN.

---

slow down the training. Furthermore, since finding the appropriate filters for their elimination is NP-hard, certain methods resorted to selecting filters based on their importance [47, 44, 46], or via genetic [27] or greedy [30] algorithms.

**Redundancy elimination** Approaches, including Sauron, that identify redundant filters by computing a similarity metric among all [43, 39] or within clusters of filters/feature maps [13, 9, 4]. Previously, cluster-based approaches have considered redundant those within-cluster filters near the Euclidean center [9] and median [13], or filters with similar $L_1$ norm over several training epochs [4]. A disadvantage of these approaches is an extra "number of clusters" hyperparameter, which is data dependent and the same hyperparameter value might not be optimal across layers. Other methods have used Pearson's correlation between the weights [43] or between the feature maps [39] within the same layer, and feature maps' rank [25] to indicate redundancy, although, their computations are more expensive than utilizing distances as in cluster-based methods.

## 3 Sauron

In this section, we present our approach to filter pruning, which we call **S**imple **AU**tomated **R**edundancy eliminati**ON** (Sauron). Sauron optimizes, jointly with the loss function, a regularization term that leads to clusters of feature maps at each convolutional layer, accentuating the redundancy of CNNs. It then eliminates the filters corresponding to the redundant feature maps by using automatically-adjusted layer-specific thresholds. Sauron requires minimal changes from the typical neural network optimization since it prunes and optimizes CNNs jointly, i.e., training involves the usual forward-backward passes and a pruning step after each epoch. Moreover, Sauron does not integrate optimizable parameters, such as channel importance masks [6, 29, 15, 26, 16], into the CNN architecture. This avoids complicating the optimization task and increasing the training time and the required GPU memory. Algorithm 1 summarizes our method.

### 3.1 Preliminaries

Let $\mathcal{D} = \{\mathbf{x}_i, \boldsymbol{y}_i\}_{i=1}^N$ represent the training set, where $\mathbf{x}_i$ denotes image $i$, $\mathbf{y}_i$ its corresponding segmentation, and $N$ is the number of images. Let $\mathbf{W}^l \in \mathbb{R}^{s_{l+1} \times s_l \times k \times k}$ be the weights, composed

by $s_{l+1}s_l$ filters of size $k \times k$ at layer $l$, where $s_{l+1}$ denotes the number of output channels, $s_l$ the number of input channels, and $k$ is the kernel size. Given feature maps $\mathbf{O}^l \in \mathbb{R}^{s_l \times h \times w}$ of $h \times w$ image dimensions, the feature maps $\mathbf{O}^{l+1} \in \mathbb{R}^{s_{l+1} \times h \times w}$ at the next layer are computed as

$$\mathbf{O}^{l+1} = \sigma(Norm(\mathbf{W}^l * \mathbf{O}^l)), \tag{1}$$

where * is the convolution operation, $Norm$ is a normalization layer, and $\sigma$ is an activation function. For simplicity, we omit the bias term in Eq. (1), and we include all CNN's parameters in $\boldsymbol{\theta} = \{\boldsymbol{W}^1, \ldots, \boldsymbol{W}^L\}$, where $L$ is the number of layers. We denote the predicted segmentation of the image $\mathbf{x}_i$ by $\hat{\boldsymbol{y}}_i$.

## 3.2 Forward pass

Sauron minimizes a loss $\mathcal{L}$ consisting of Cross Entropy $\mathcal{L}_{CE}$, Dice loss $\mathcal{L}_{Dice}$ [31], and a novel channel distance regularization term $\delta_{opt}$: $\mathcal{L} = \mathcal{L}_{CE} + \mathcal{L}_{Dice} + \lambda\delta_{opt}$, where

$$\delta_{opt} = \frac{1}{L} \sum_{l=1}^{L} \frac{1}{s_{l+1}} \sum_{r=2}^{s_{l+1}} ||\phi(\boldsymbol{O}_1^l; \omega) - \phi(\boldsymbol{O}_r^l; \omega)||_2, \tag{2}$$

$\lambda$ is a hyperparameter that balances the contribution of $\delta_{opt}$, and $\phi$ denotes average pooling with window size and strides $\omega$. Before computing $\delta_{opt}$, feature maps $\boldsymbol{O}_1^l$ and $\boldsymbol{O}_{-1}^l$ (all channels except the first) are normalized to the range $[0, 1]$ via min-max normalization, as we experimentally found this normalization strategy to be the best (see Appendix A). For pruning, Sauron computes distances between a randomly-chosen feature map $\pi \in \{1, \ldots, s_{l+1}\}$ and all the others: $\boldsymbol{\delta}_{prune} = \{d_r^l / \max_r d_r^l : l = 1, \ldots, L, r = 1, \ldots, \pi - 1, \pi + 1, \ldots, s_{l+1}\}$, where

$$d_r^l = ||\phi(\boldsymbol{O}_\pi^l; \omega) - \phi(\boldsymbol{O}_r^l; \omega)||_2. \tag{3}$$

Importantly, $\pi$ is different in every layer and epoch, enabling Sauron to prune different feature map clusters. Moreover, since finding an appropriate pruning threshold requires the distances to lie within a known range, Sauron normalizes $d_r^l$ such that their maximum is 1, i.e., $d_r^l \leftarrow d_r^l / \max_r(d_r^l)$.

## 3.3 Backward pass: $\delta_{opt}$ regularization

Optimized CNNs have been shown to have redundant weights and to produce redundant feature maps [13, 43] (Appendix E). By minimizing the extra regularization term $\delta_{opt}$, CNNs further promote the formation of clusters, facilitating their subsequent pruning. $\delta_{opt}$ regularization makes those feature maps near the feature map in the first channel $\boldsymbol{O}_1^l$ (i.e., within the same cluster) even closer. At the same time, those feature maps that are dissimilar to $\boldsymbol{O}_1^l$ (i.e., in other clusters) become more similar to other feature maps from the same cluster, as it holds that $||\phi(\boldsymbol{O}_i^l; \omega) - \phi(\boldsymbol{O}_j^l; \omega)||_2 \leq ||\phi(\boldsymbol{O}_1^l; \omega) - \phi(\boldsymbol{O}_i^l; \omega)||_2 + ||\phi(\boldsymbol{O}_1^l; \omega) - \phi(\boldsymbol{O}_j^l; \omega)||_2$ for $i \neq j$, i.e., the right hand side—minimized via $\delta_{opt}$ regularization—is an upper bound of the left hand side. We demonstrate this clustering effect in Section 4.2. Furthermore, for pruning, we focus on the feature maps rather than on the weights since different non-redundant weights can lead to similar feature maps. Thus, eliminating redundant weights guarantees no reduction in feature maps redundancy.

## 3.4 Pruning step

Sauron employs layer-specific thresholds $\boldsymbol{\tau} = [\tau_1, \ldots, \tau_L]$, where all $\tau_l$ are initialized to zero and increase independently (usually at a different pace) until reaching $\tau_{max}$. This versatility is important as the ideal pruning rate differs across layers due to their different purpose (i.e., extraction of low- and high-level features) and their varied number of filters. Additionally, this setup permits utilizing high thresholds without removing too many filters at the beginning of the optimization, as feature maps may initially lie close to each other due to the random initialization. In consequence, pruning is embedded into the training and remains *always active*, portraying Sauron as a single-phase filter pruning method.

**Procedure 1: Increasing $\tau_l$**    Pruning with adaptively increasing layer-specific thresholds raises two important questions: how and when to increase the thresholds? Sauron increases the thresholds linearly in $\kappa$ steps until reaching $\tau_{max}$. Then, thresholds are updated once the model has stopped improving (C1 and C2 in Algorithm 1) and it has pruned only a few filters (C3). An additional "patience" hyperparameter ensures that the thresholds are not updated consecutively (C4). Conditions C1,...,C4 are easy to implement and interpret, and they rely on heuristics commonly employed for detecting convergence.

**Procedure 2: Pruning**    Sauron considers nearby feature maps to be redundant since they likely belong to the same cluster. In consequence, Sauron removes all input filters $\mathbf{W}^l_{\cdot, s_l}$ whose corresponding feature map distances $\boldsymbol{\delta}_{prune}$ are lower than threshold $\tau_l$. In contrast to other filter pruning methods, Sauron needs to store no additional information, such as channel indices, and the pruned models become more efficient *and* smaller. Additionally, since pruning occurs during training, Sauron accelerates the optimization of CNNs. After training, pruned models can be easily loaded by specifying the new post-pruning number of input and output filters in the convolutional layers.

### 3.5 Implementation

Sauron's simple design permits its incorporation into existing CNN optimization frameworks easily. As an example, in our implementation, convolutional blocks are wrapped into a class that computes $\delta_{opt}$ and $\boldsymbol{\delta}_{prune}$ effortlessly in the forward pass, and the pruning step is a callback function triggered after each epoch. This implementation, together with the code for running our experiments and processing the datasets, was written in Pytorch [33] and is publicly available at `https://github.com/blindedrepository`. In our experiments, we utilized an Nvidia GeForce GTX 1080 Ti (11GB), and a server with eight Nvidia A100 (40GB).

## 4    Experiments

In this section, we compare Sauron with other state-of-the-art filter pruning methods and conduct an ablation study to show the impact on pruning and performance of $\delta_{opt}$ regularization. We empirically demonstrate that the proposed $\delta_{opt}$ regularization increases feature map clusterability, and we visualize the feature maps of a Sauron-pruned model.

**Datasets**    We employed three 3D medical image segmentation datasets: Rats, ACDC, and KiTS. *Rats* comprised 160 3D T2-weighted magnetic resonance images of rat brains with lesions [41], and the segmentation task was separating lesion from non-lesion voxels. We divided Rats dataset into 0.8:0.2 train-test splits, and the training set was further divided into a 0.9:0.1 train-validation split, resulting in 115, 13, and 32 images for training, validation, and test, respectively. *ACDC* included the Automated Cardiac Diagnosis Challenge 2017 training set [5] (CC BY-NC-SA 4.0), comprised by 200 3D magnetic resonance images of 100 individuals. The segmentation classes were background, right ventricle (RV), myocardium (M), and left ventricle (LV). We divided ACDC dataset similarly to Rats dataset, resulting in 144, 16, and 40 images for training, validation, and test, respectively. We only utilized ACDC's competition training set due to the limitation to only four submissions to the online platform of ACDC challenge. Finally, *KiTS* was composed by 210 3D images from Kidney Tumor Challenge 2019 training set, segmented into background, kidney and kidney tumor [14] (MIT). KiTS training set was divided into a 0.9:0.1 train-validation split, resulting in 183 and 21 images for training and validation. We report the results on the KiTS's competition test set (90 3D images). All 3D images were standardized to zero mean and unit variance. The train-validation-test divisions and computation of the evaluation criteria was at the subject level, ensuring that the data from a single subject was completely in the train set or in the test set, never dividing subject's data between train and test sets. See Appendix C for preprocessing details.

**Model and optimization**    Sauron and the compared filter pruning methods optimized nnUNet [18] via deep supervision [22] with Adam [20] starting with a learning rate of $10^{-3}$, polynomial learning

Table 1: Performance on Rats dataset.

Table 2: Performance on ACDC dataset. **Bold**: best performance among pruning methods.

| Method | Lesion | | LV | | M | | RV | |
|---|---|---|---|---|---|---|---|---|
| | Dice | HD95 | Dice | HD95 | Dice | HD95 | Dice | HD95 |
| nnUNet | 0.94 $\pm$ 0.03 | 1.1 $\pm$ 0.3 | 0.91 $\pm$ 0.05 | 4.4 $\pm$ 3.0 | 0.90 $\pm$ 0.02 | 3.4 $\pm$ 5.8 | 0.95 $\pm$ 0.03 | 2.5 $\pm$ 1.8 |
| Sauron | **0.94** $\pm$ **0.03** | 1.1 $\pm$ 0.3 | **0.90** $\pm$ **0.06** | **4.7** $\pm$ **3.2** | **0.90** $\pm$ **0.02** | 3.6 $\pm$ 8.0 | **0.95** $\pm$ **0.03** | **2.7** $\pm$ **2.0** |
| Sauron ($\lambda = 0$) | 0.93 $\pm$ 0.03 | 1.2 $\pm$ 0.5 | 0.89 $\pm$ 0.08 | 5.3 $\pm$ 4.4 | **0.90** $\pm$ **0.02** | **2.4** $\pm$ **1.7** | **0.95** $\pm$ **0.03** | 3.1 $\pm$ 3.0 |
| cSGD ($r = 0.5$) | 0.86 $\pm$ 0.13 | 9.6 $\pm$ 16.8 | 0.10 $\pm$ 0.15 | 72.6 $\pm$ 74.1 | 0.54 $\pm$ 0.19 | 19.5 $\pm$ 35.6 | 0.64 $\pm$ 0.20 | 13.9 $\pm$ 8.2 |
| FPGM ($r = 0.5$) | 0.93 $\pm$ 0.04 | **0.5** $\pm$ **0.5** | 0.57 $\pm$ 0.13 | 37.8 $\pm$ 7.3 | 0.89 $\pm$ 0.03 | 2.2 $\pm$ 1.6 | 0.00 $\pm$ 0.00 | 194.1 $\pm$ 23.5 |
| Autopruner | 0.91 $\pm$ 0.04 | 0.8 $\pm$ 1.2 | 0.88 $\pm$ 0.07 | 5.9 $\pm$ 4.6 | 0.88 $\pm$ 0.03 | 2.5 $\pm$ 1.7 | **0.95** $\pm$ **0.03** | 3.1 $\pm$ 3.0 |

Table 3: Performance on KiTS datasets.

| Method | Kidney | Tumor |
|---|---|---|
| | Dice | Dice |
| nnUNet [17] | 0.9595 | 0.7657 |
| Sauron | **0.9564** | **0.7482** |
| Sauron ($\lambda = 0$) | 0.9556 | 0.7352 |
| cSGD [9] ($r = 0.5$) | 0.9047 | 0.5207 |
| FPGM [13] ($r = 0.5$) | 0.9509 | 0.6830 |
| Autopruner [29] | 0.9167 | 0.5854 |

Table 4: Decrease in FLOPs with respect to the baseline nnUNet. **Bold**: highest decrease.

| Method | Rats | ACDC | KiTS |
|---|---|---|---|
| nnUNet [17] | 0.00% | 0.00% | 0.00% |
| Sauron | 96.45% | **92.41%** | **93.02%** |
| Sauron ($\lambda = 0$) | **96.62%** | 89.04% | 85.82% |
| cSGD [9] ($r = 0.5$) | 50.03% | 49.80% | 49.81% |
| FPGM [13] ($r = 0.5$) | 50.00% | 50.0% | 49.98% |
| Autopruner [29] | 83.61% | 88.52% | 82.00% |

rate decay, and weight decay of $10^{-5}$. During training, images were augmented with TorchIO [34] (see Appendix C). nnUNet is a self-configurable U-Net and the dataset optimized nnUNet architectures slightly differed on the number of filters, encoder-decoder levels, normalization layer, batch size, and number of epochs (see Appendix C).

**Pruning** Sauron decreased feature maps dimensionality via average pooling with window size and stride of $\omega = 2$, and utilized $\lambda = 0.5$ in the loss function, maximum pruning threshold $\tau_{max} = 0.3$, pruning steps $\kappa = 15$, and patience $\rho = 5$ (C4 in Algorithm 1). Additionally, we employed simple conditions to detect convergence for increasing the layer-specific thresholds $\tau$. Convergence in the training loss (C1) was detected once the most recent training loss lay between the maximum and minimum values obtained during the training. We considered that the validation loss stopped improving (C2) once its most recent value increased with respect to all previous values. Finally, the remaining condition (C3) held true if the layer-specific threshold pruned less than 2% of the filters pruned in the previous epoch, i.e., $\mu = 2$.

## 4.1 Benchmark on three segmentation tasks

We optimized and pruned nnUNet [18] with Sauron, and we compared its performance with cSGD[1] [9], FPGM[2] [13], and Autopruner[3] [29] using a pruning rate similar to the one achieved by Sauron. Since cSGD and FPGM severely underperformed in this setting, we re-run them with their pruning rate set to only 50% ($r = 0.5$). Additionally, to understand the influence of the proposed regularization term $\delta_{opt}$ on the performance and pruning rate, we conducted ablation experiments with $\lambda = 0$. We computed the Dice coefficient [8] and 95% Hausdorff distance (HD95) [37] on Rats and ACDC test sets (see Tables 1 and 2). In KiTS dataset, only the average Dice coefficient was provided by the online platform that evaluated the test set (see Table 3). In addition to Dice and HD95, we computed the relative decrease in the number of floating point operations (FLOPs) in all convolutions: $FLOPs = HW(C_{in}C_{out})K^2$, where $H, W$ is the height and width of the feature maps, $C_{in}, C_{out}$ is the number of input and output channels, and $K$ is the kernel size. For the 3D CNNs (KiTS dataset), an extra $D$ (depth) and $K$ are multiplied to compute the FLOPs.

Sauron obtained the highest Dice coefficients and competitive HD95s across all datasets and segmentation classes (Tables 1 to 3). Sauron also achieved the highest reduction in FLOPs, although, every

---

[1] https://github.com/DingXiaoH/Centripetal-SGD
[2] https://github.com/he-y/filter-pruning-geometric-median
[3] https://github.com/Roll920/AutoPruner

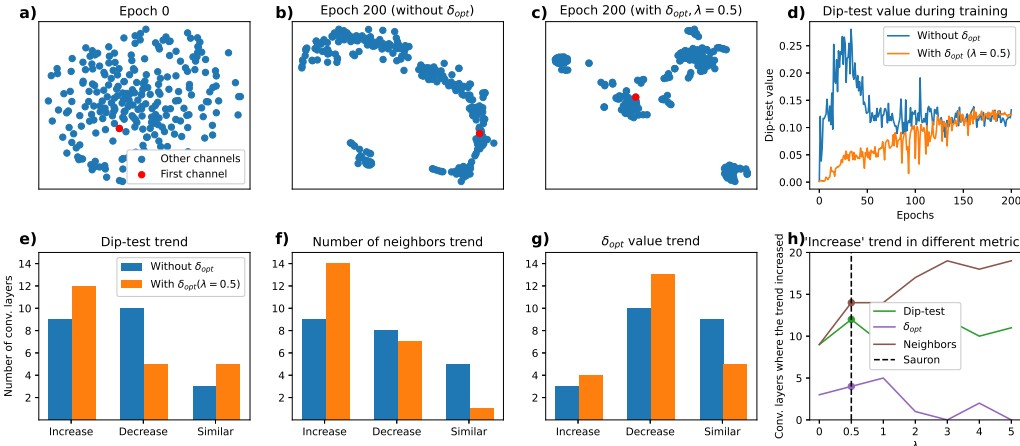

Figure 1: a-c) tSNE plot of "*dec_block_1*" feature maps at initialization (epoch 0), and after optimizing with and without $\delta_{opt}$. d) Corresponding dip-test values during the optimization. e-g) Summary of the trends across the three clusterability measures in all convolutional layers. h) Number of layers with an increasing trend in the three clusterability measures with higher values of $\lambda$ (dashed line: Sauron's default configuration).

method, including Sauron, can further reduce the FLOPs at the risk of worsening the performance (Table 4). cSGD and FPGM could not yield models with high pruning rates possibly because they aim at reducing only $s_{l+1}$ and not $s_l$ from $\mathbf{W}^l \in \mathbb{R}^{s_{l+1} \times s_l \times k \times k}$. Thus, very high pruning rates cause a great imbalance between the number of input and output filters in every layer that may hinder the training. Note also that cSGD and FPGM were not tested with pruning rates higher than 60% [9, 13]. In contrast, Sauron and Autopruner that achieved working models with higher pruning rate reduced both input filters $s_l$ and output filters $s_{l+1}$.

Sauron without the proposed regularization term $\delta_{opt}$ (Sauron ($\lambda = 0$)) achieved similar or less compressed models and worse Dice coefficients than when minimizing $\delta_{opt}$. Overall, the results from these ablation experiments indicate that 1) typical CNN optimization (without $\delta_{opt}$ regularization) yields redundant feature maps that can be pruned with Sauron, 2) pruning rate is generally higher with $\delta_{opt}$ regularization, and 3) pruning with no $\delta_{opt}$ regularization can affect performance, possibly due to the accidental elimination of non-redundant filters. In summary, the pruning rate and performance achieved in our ablation experiments demonstrate that promoting clusterability via $\delta_{opt}$ regularization is advantageous for eliminating redundant feature maps.

### 4.2   Minimizing $\delta_{opt}$ promotes the formation of feature maps clusters

We investigated feature map clustering tendency during nnUNet's optimization. For this, we deactivated Sauron's pruning step and optimized $\mathcal{L}$ on Rats dataset with and without $\delta_{opt}$ while storing the feature maps at each epoch (including at epoch 0, before the optimization) of every convolutional layer. Since quantifying clusterability is a hard task, we utilized three different measures: 1) We employed **dip-test** [19], as Adolfsson et al. [2] demonstrated its robustness compared to other methods for quantifying clusterability. High dip-test values signal higher clusterability. 2) We computed the average **number of neighbors** of each feature map layer-wise. Specifically, we counted the feature maps within $r$, where $r$ corresponded to the 20% of the distance between the first channel and the farthest channel. Distance $r$ is computed every time since the initial distance between feature maps is typically reduced while training. An increase in the average number of neighbors indicates that feature maps have become more clustered. 3) We calculated the **average distance** to the first feature map channel (i.e., $\delta_{opt}$) for each layer, which illustrates the total reduction of those distances achieved during and after the optimization.

In agreement with the literature [13, 43], Figure 1 shows that optimizing nnUNet (without $\delta_{opt}$ regularization) yields clusters of feature maps. Feature maps in layer "*dec_block_1*" (see Appendix B) show no apparent structure suitable for clustering at initialization (Fig. 1, a), and, at the end of the optimization, feature maps appear more clustered (Fig. 1, b). Figure 1 (d, blue line) also illustrates this phenomenon: dip-test value is low in the beginning and higher at the end of the training. However, this increasing trend did not occur in all layers. To illustrate this, we compared, for each layer, the average dip-test value, number of neighbors, and distance $\delta_{opt}$ in the first and last third of the training. Then, we considered the trend similar if the difference between these values was smaller than 0.001 (for the dip-test values) or smaller than 5% of the average value in the first third (for the number of neighbors and distance $\delta_{opt}$). Figure 1 (e) shows that the number of layers in which the dip-test value increased and decreased were similar when not minimizing the $\delta_{opt}$ regularization term. In contrast, the number of layers with an increasing trend was proportionally larger with $\delta_{opt}$ regularization. Figure 1 (f) shows a similar outcome regarding the average number of neighbors, i.e., $\delta_{opt}$ regularization led to proportionally more neighbors near each feature map. In the same line, the average distance between the first feature map and the rest decreased more with $\delta_{opt}$ regularization (Fig. 1, (f)). Additionally, Figure 1 (c) also illustrates that incorporating the $\delta_{opt}$ regularization term enhances the clustering of feature maps, as there are more clusters and the feature maps are more clustered than when not minimizing $\delta_{opt}$ (Fig. 1 (b)).

We observed higher clusterability in the convolutional layers with more feature maps (see Appendix D). This is likely because such convolutional layers contribute more to the value of $\delta_{opt}$ (Eq. 2). On the other hand, convolutional layers with fewer feature maps have larger feature vectors (e.g., *enc_block_1* feature vectors are $(256 \times 256) \times 32$ in Rats dataset) whose distances tend to be larger due to the curse of dimensionality. Sauron accounts, to some extent, for these differences in the convolutional layers with the adaptively-increasing layer-specific thresholds $\tau$. Another possible way to tackle these differences is by using different layer-specific $\lambda$'s to increase the contribution of the distances of certain layers. We investigated the impact on feature map clusterability with higher $\lambda$ values and, as illustrated in Figure 1 (h), a higher $\lambda$ tended to increase the average number of neighbors, decrease $\delta_{opt}$, and somewhat increase the dip-test values, which, overall, signals higher clusterability.

## 4.3 Feature maps interpretation

Sauron produces small and efficient models that can be easier to interpret. This is due to $\delta_{opt}$ regularization that, as we showed in Section 4.2, increases feature maps clusterability. Each feature maps cluster can be thought of as a semantic operation and the cluster's feature maps as noisy outputs of such operation. To test this view, we inspected the feature maps from the second-to-last convolutional block (*dec_block_8*, see Appendix B) of a Sauron-pruned nnUNet. For comparison, we included the feature maps from the same convolutional layer of the baseline (unpruned) nnUNet in Appendix E.

The first feature map depicted in Figure 2 (top) captured the background and part of the rat head that does not contain brain tissue. The second feature map contained the rest of the rat head without brain lesion, and the third feature map mostly extracted the brain lesion. Although the third feature map seems to suffice for segmenting the brain lesion, the first feature map might have helped the model by discarding the region with no brain tissue at all. Similarly, the first and second feature maps in Figure 2 (middle) detected the background, whereas feature maps 3, 4, and 5 extracted, with different intensities, the right cavity (red), myocardium (green), and left cavity (blue) of the heart. In Figure 2 (bottom), we can also see that each feature map captured the background, kidney (red), and tumor (blue) with different intensities. This high-level interpretation facilitates understanding the role of the last convolutional block which, in the illustrated cases, could be replaced by simple binary operations. This shows the interpretability potential of feature map redundancy elimination methods such as Sauron.

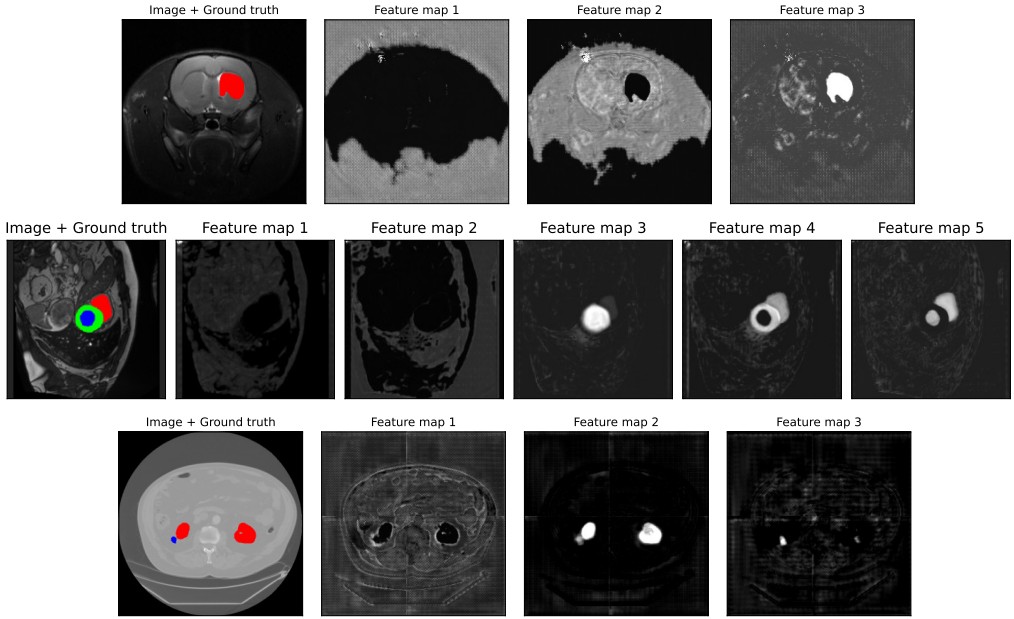

Figure 2: Image slice from Rats (top), ACDC (middle), and KiTS (bottom) datasets, its ground-truth segmentation, and all feature maps at the second-to-last convolutional block after pruning with Sauron.

## 5   Conclusion

We presented our single-phase filter pruning method named Sauron, and we evaluated it on three medical image segmentation tasks in which Sauron yielded pruned models that were superior to the compared methods in terms of performance and pruning rate. In agreement with the literature, our experiments indicated that CNN optimization leads to redundant feature maps that can be clustered. Additionally, we introduced Sauron's $\delta_{opt}$ regularization that, as we showed with three different clusterability metrics, increased feature maps clusterability without pre-selecting the number of clusters, unlike previous approaches. In other words, we enhanced CNN's innate capability to yield feature maps clusters via $\delta_{opt}$ regularization, and we exploited it for filter pruning. Finally, we showed that the few feature maps after pruning nnUNet with Sauron were highly interpretable.

**Limitations and potential negative impact**    Sauron relies on feature maps for identifying which filters to prune. Thus, although Sauron is suitable for training models from scratch and fine-tuning pre-trained networks, Sauron is unable to prune CNNs without access to training data, unlike [23, 43, 24]. Furthermore, Sauron cannot enforce a specific compression rate due to its simple distance thresholding. Although we have evaluated Sauron with respect to the segmentation quality, we are not able to evaluate the potential clinical impact. It could be that even a small difference in segmentation would have large clinical impact, or vice versa, a large difference in segmentation could be clinically meaningless. Depending on the application these impacts could be either positive or negative.

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
