# OpenReview forum: "Sauron U-Net: Simple automated redundancy elimination in medical image segmentation via filter pruning"
_NeurIPS.cc/2022/Conference — NeurIPS 2022 Submitted_

### Official Review · Reviewer_sPLR · 2022-07-08

**Rating:** 4
**Confidence:** 4
**Soundness:** 2 fair
**Presentation:** 3 good
**Contribution:** 1 poor

**Summary:**

The authors proposed Sauron, a filter pruning methodology designed for U-Net-like medical image segmentation networks. In comparison to most filter pruning frameworks that consist of more than one distinct phase, Sauron applies filter pruning during optimization in a single phase, making it require fewer hyperparameters and design decisions. To achieve this goal, Sauron facilitates and promotes the formation of feature map clusters by optimizing a regularization term, and does not enforce the number of these clusters. In experiments, Sauron achieves comparable segmentation performance and largely reduced FLOPs on nnUNet.

**Questions:**

1. Lines 29-30: The statement "models with a few filters can be easier to interpret than large models, which is crucial not only in clinical applications but also in research." needs citations.

2. Table 4 is never referred in the main text.

3. The provided github link does not exist.

**Ethics Review Area:**

["I don’t know"]

**Limitations:**

The authors adequately addressed the limitations and potential negative societal impact of their work.

**Strengths And Weaknesses:**

Strengths:

1. The proposed clustering-based filter pruning method sounds interesting and reasonable. The regularization term $\delta_{opt}$ in Sauron makes those feature maps near the feature map in the first channel even closer. Meanwhile, based on the upper bound in lines 129-130, $\delta_{opt}$ forces those feature maps that are dissimilar to the feature map in the first channel to be more similar to other feature maps from the same cluster.

2. Sauron performs well on 2D nnUNet, providing comparable performance on Rats and ACDC datasets with largely reduced FLOPs.

Weaknesses:

1. In Tables 1-3, it seems that the regularization term $\delta_{opt}$ does not have a significant impact on the segmentation performance. For example, Sauron $\delta_{opt}=0$ performs comparably with Sauron on Rats, ACDC, and KiTS (Kidney). In my opinion, I think these results cannot fully reflect the significance of the introduced regularization term (the major contribution of Sauron).

2. The results of 3D nnUNet on KiTS are not satisfactory, especially on the Tumor class. Actually, I recommend the authors to lay more emphasize on 3D UNets, which often require many more computational resources than 2D UNets. However, most of the experiments were made on 2D UNets, which are not deep and maintain high inference speed (not really in need of filter pruning).

3. The authors stated that Sauron is a single phase pruning method, which is more advantageous since it requires fewer hyperparameters and design decisions, including the number of epochs for training and fine-tuning, pruning iterations, etc. The authors should verify their statements with either experimental results or appropriate citations.

4. The pruning baselines, i.e., cSGD, FPGM, and Autopruner, are somewhat out-of-dated, which were published about three years ago. The authors should consider including more recent (published within two years) filter pruning approaches for fair comparisons.

---

> ### Author Response · Authors · 2022-08-02
> **Response**
>
> We really appreciate the feedback given and thank you for your time and useful comments.
>
> 1 - $\delta_{opt}$ regularization term was designed to increase redundancy in the feature maps to facilitate the subsequent filter pruning. In other words, minimizing $\delta_{opt}$ was not intended to increase performance directly unlike perhaps other types of regularization such as L2 regularization. To this end, we showed that minimizing $\delta_{opt}$ did increase the clusterability of feature maps (Section 4.2) and that the pruned models became smaller than without $\delta_{opt}$ regularization (Table 4). Furthermore, in our experiments, we observed a slight increase in performance when minimizing $\delta_{opt}$.
>
> 2 - We employed 2D and 3D UNets as both types are widely used in medical image segmentation. Particularly, 2D UNets are often the preferred choice in 3D anisotropic data (see e.g. De Feo et al., 2021), such as in Rats and ACDC datasets, or in 2D medical images.
>
> The performance on KiTS test set was obtained by training the architecture that won KiTS19 competition (nnUNet) following a very similar configuration (reported in the supplementary material). We agree with the reviewer that a higher performance would be more satisfactory, but our focus was not on surpassing a particular metric in KiTS19 test set, which would have required a very careful choice in the preprocessing, data augmentation, and postprocessing steps. Instead, our focus was on achieving a higher pruning rate while not decreasing performance compared to other filter pruning methods.
>
> De Feo R, Shatillo A, Sierra A, Valverde JM, Gröhn O, Giove F, Tohka J. “Automated joint skull-stripping and segmentation with Multi-Task U-Net in large mouse brain MRI databases”. NeuroImage. 2021 Apr 1;229:117734.
>
> 3 - We thank the reviewer for this suggestion and we will include an appropriate citation. In agreement with Huang and Wang (2018), we believe that fewer hyperparameters and fewer design decisions (as single-phase pruning methods) are positive and desirable properties of pruning approaches.
>
> Huang Z, Wang N. “Data-driven sparse structure selection for deep neural networks”. In Proceedings of the European conference on computer vision (ECCV) 2018 (pp. 304-320).
>
> 4 - We think that the compared methods (cSGD, FPGM, Autopruner) are still state-of-the-art; they were published within two years since we started our work, are widely used, and they have been compared with filter pruning methods published this year (e.g., Wang and Li (2022), Li et al. (2022), Joo et al. (2022), Yu et al. (2022), Hou et al. (2022)). These compared methods were chosen based on the availability of code and the similarity to our approach.
>
> Wang Z, Li C. “Channel Pruning via Lookahead Search Guided Reinforcement Learning”. In Proceedings of the IEEE/CVF Winter Conference on Applications of Computer Vision 2022 (pp. 2029-2040).
>
> Li Y, Adamczewski K, Li W, Gu S, Timofte R, Van Gool L. “Revisiting Random Channel Pruning for Neural Network Compression”. In Proceedings of the IEEE/CVF Conference on Computer Vision and Pattern Recognition 2022 (pp. 191-201).
>
> Joo D, Kim D, Yi E, Kim J. “Linear Combination Approximation of Feature for Channel Pruning”. In Proceedings of the IEEE/CVF Conference on Computer Vision and Pattern Recognition 2022 (pp. 2772-2781).
>
> Yu S, Mazaheri A, Jannesari A. “Topology-Aware Network Pruning using Multi-stage Graph Embedding and Reinforcement Learning”. In International Conference on Machine Learning 2022 Jun 28 (pp. 25656-25667). PMLR.
>
> Hou Z, Kung SY. “Multi-dimensional dynamic model compression for efficient image super-resolution”. In Proceedings of the IEEE/CVF Winter Conference on Applications of Computer Vision 2022 (pp. 633-643).
>
> **Questions**
>
> 1 - We agree with the reviewer. We will cite the work of Reyes et al. (2020) that connects models' large number of parameters and models’ interpretability in the context of medical imaging.
>
> Reyes M, Meier R, Pereira S, Silva CA, Dahlweid FM, von Tengg-Kobligk H, Summers RM, Wiest R. “On the interpretability of artificial intelligence in radiology: challenges and opportunities”. Radiology: artificial intelligence. 2020 May;2(3).
>
> 2 - We referred to Table 4 in line 221.
>
> 3 - The provided GitHub link was a placeholder to be replaced by the official GitHub repository link upon acceptance. We included the content of that GitHub repository as the Supplementary Material (including Sauron’s code and a README file that describes how to run all our experiments).

---

> > ### Comment · Reviewer_sPLR · 2022-08-05
> > **Two concerns still remain**
> >
> > 1. $\delta_{opt}$ is listed among the top-2 contributions in the submission. However, from Tables 1-4, I did not get the significance of $\delta_{opt}$ in either segmentation or FLOPs. Then, what is the exact meaning of $\delta_{opt}$?
> >
> > 2. More experiments using 3D UNet/nnUNet are necessary. In my view, reducing the FLOPs of 2D UNet is not that important because in most cases, we do not need to run medical diagnosis on a mobile device (we can run it on a server).

---

> > > ### Author Response · Authors · 2022-08-08
> > > **Response**
> > >
> > > We thank you for reading and answering our response, and we hope to address the two remaining concerns.
> > >
> > > 1- The significance of $\delta_{opt}$ regularization is in the greater reduction of FLOPs without harming performance. This is possible because $\delta_{opt}$ regularization further promotes feature maps clustering, as we demonstrate in Section 4.2. In other words, pruning without $\delta_{opt}$ regularization ($\lambda=0$) may either not eliminate enough feature maps because feature maps are less clustered, or it may accidentally remove feature maps that are not redundant (line 231).
> > >
> > > Specifically, Tables 1-3 (second vs. third row) show that our $\delta_{opt}$ regularization did not deteriorate Dice coefficient and HD95, and Table 4 (second vs. third row) shows that $\delta_{opt}$ regularization resulted in a higher reduction in the FLOPs in ACDC and KiTS datasets. In Rats dataset, however, given the smaller performance without $\delta_{opt}$ regularization (Table 1, second vs. third row), the marginally higher reduction in FLOPs could indicate the accidental elimination of non-redundant feature maps, as aforementioned (line 231).
> > >
> > > 2- We agree that more experiments on 3D networks would be beneficial in further illustrating the effectiveness of Sauron. We employed 2D and 3D CNNs on three distinct datasets (Section 4.1), and, in our experiments with 3D CNNs (KiTS dataset), Sauron outperformed, by a large margin, the competing methods, achieving higher Dice coefficients (Table 3) and greater FLOPs decrease (Table 4, last column). Given these results, and since Sauron focuses on increasing the redundancy across feature maps regardless of whether convolutional filters are 2D or 3D kernels, we expect that Sauron will perform favorably on other 3D CNN architectures.
> > >
> > > We think that reducing the FLOPs on 2D UNets is also important. Certain 2D medical and biomedical images, such as those from cancer pathology sections and those obtained in electron microscopy, are comprised by millions of voxels (e.g., A.B. Hamida et al. (2021) employed the AiCOLO dataset in which each image encompasses around 4 to 5 billion pixels, S. Akers et al. (2021) utilized 3042x3044 pixel images). Thus, achieving small 2D UNets enables efficient whole-volume segmentation in such cases. Additionally, pruning 2D UNets during the optimization (as Sauron does) decreases gradually GPU memory usage, permitting to increase the batch size, which, in medical image segmentation, is typically much smaller than in natural image segmentation. Furthermore, due to patient privacy issues, it is not always possible to run analyses on servers. We think that our experiments in both 2D and 3D networks (Section 4.1) strengthen our work as they showed that Sauron can prune different types of networks (2D and 3D) and architectures (Tables 1-3, Appendix C).
> > >
> > > Akers, Sarah, Elizabeth Kautz, Andrea Trevino-Gavito, Matthew Olszta, Bethany E. Matthews, Le Wang, Yingge Du, and Steven R. Spurgeon. "Rapid and flexible segmentation of electron microscopy data using few-shot machine learning." npj Computational Materials 7, no. 1 (2021): 1-9.
> > > Hamida, A. B., Devanne, M., Weber, J., Truntzer, C., Derangère, V., Ghiringhelli, F., Forestier, G. & Wemmert, C. (2021). Deep learning for colon cancer histopathological images analysis. Computers in Biology and Medicine, 136, 104730.

---

### Official Review · Reviewer_xHRW · 2022-07-08

**Rating:** 5
**Confidence:** 4
**Soundness:** 3 good
**Presentation:** 3 good
**Contribution:** 3 good

**Summary:**

The paper proposed a method for pruning filters in image segmentation networks by removing filters during training that are closely clustered. Unlike prior works, the approach is described as single-phase, meaning it prunes during normal training. To promote smaller networks, a term which promotes feature map clustering is added to the loss. The experiments uses nnUNet as a baseline network and 3 other recent network pruning methods as comparison. The results show better performance (Dice/HD95) with more pruned networks.

**Questions:**

- It seems like the proposed method does have a number of additional hyperparameters. See for instance lambda (l114), the layer-specific thresholds (l137), the patience hyperparameter (l149), the number of pruning steps (l198), etc. Considering that more hyperparameters are mentioned as a weakness of the prior works in contrast to the proposed, the authors should maybe clarify why and if they consider their approach superior in this respect?


**Limitations:**

- Aside from the above limitations, I think the work addresses limitations as aspected.


**Strengths And Weaknesses:**

+ Well motivated.

+ clearly written and easy to follow.

- I find significance is hard to asses, as the results and the novel contributions of the present work is no
t put and discussed in the context of prior work to a high enough degree.

- Having a list of contributions at the end of introduction, I feel is a good way to summarize reasons the a particular manuscript is relevant to read. In this case, however, it repeats too much for my taste. I would
 have preferred a stronger focus on novelty and what separates this work from prior works specifically.

- The section on Previous work hard to follow. I would have preferred more detail and more relevance given the context of the submitted work. As it is, it just reads as a listing of prior approaches, with too little detail to be informative.

- The section on Feature maps interpretation could benefit greatly from having the comparison to the non-pruned nnUnet in the paper and not in the appendix. As it is, it is not possible to assess the claims that the pruning makes the models easier to interpret.

- It is unclear to me if the clustering experiments (Figure 1) were done with multiple training runs. A single experiment where clustering is improved is perhaps not so interesting.

---

> ### Author Response · Authors · 2022-08-02
> **Response**
>
> We thank you for your time and feedback provided.
>
> > * I find significance is hard to asses, as the results and the novel contributions of the present work is no t put and discussed in the context of prior work to a high enough degree.
>
> We thank the reviewer for this comment on the Previous work section. We divided previous works based on the type of pruning method in order to separate those approaches more related to our work (Redundancy elimination methods).
>
> > * The section on Feature maps interpretation could benefit greatly from having the comparison to the non-pruned nnUnet in the paper and not in the appendix. As it is, it is not possible to assess the claims that the pruning makes the models easier to interpret.
>
> We agree with the reviewer that it would be beneficial to have those figures in the paper rather than in the appendix. Those figures are quite big as they contain several feature maps. Since we wanted to show them in high resolution and due to space limitations, we placed those figures in the appendix.
>
> > * It is unclear to me if the clustering experiments (Figure 1) were done with multiple training runs. A single experiment where clustering is improved is perhaps not so interesting.
>
> The clustering experiments in Figure 1 were done on a single run. Due to the stochasticity of CNNs optimization, there is no exact correspondence between feature maps in the same layer across different runs (i.e., feature map 1 in layer $l$ might belong to different clusters in different runs). Thus, plotting "average" tSNE plots (Fig 1-a,b,c) would not have been possible. Then, to understand the evolution of feature map clustering during and after the optimization, we also computed Fig 1-d,e,f,g. (details can be found in Section 4.2). We agree with the reviewer that a single experiment where clustering is improved is not that interesting. In line with this, Fig. 1-h offers the results of this experiment with different $\lambda$ values to understand the effect of Sauron’s $\lambda$ on clustering. The trend depicted in Fig 1-h (each $\lambda$ is a different run), the rest of the analysis in Section 4.2, and the FLOPs reduction in the trained models (Table 4) show that clustering was improved via our proposed $\delta_{opt}$ regularization.
>
> > * It seems like the proposed method does have a number of additional hyperparameters. See for instance lambda (l114), the layer-specific thresholds (l137), the patience hyperparameter (l149), the number of pruning steps (l198), etc. Considering that more hyperparameters are mentioned as a weakness of the prior works in contrast to the proposed, the authors should maybe clarify why and if they consider their approach superior in this respect?
>
> Sauron has no hyperparameters that alter the typical flow of the optimization in CNNs, which, in practice, is advantageous because it facilitates the implementation of our method and the tuning of Sauron's hyperparameters. Specifically, we can divide the hyperparameters of filter pruning methods into two groups: 1) hyperparameters that affect the way in which pruning is performed (e.g. Sauron's hyperparameters, line 1 in Algorithm 1), and 2) hyperparameters related to the design decisions of the general pruning strategy, such as "number of epochs before pruning starts" and "pruning iterations". These two types of hyperparameters are highly coupled. For instance, the hyperparameter "number of epochs before pruning starts" can easily depend on how challenging a segmentation task is, and based on this hyperparameter's value, pruning hyperparameters, such as Sauron's $\tau_{max}$ (maximum threshold) or $\rho$ (patience) would differ. Thus, filter pruning methods that have no hyperparameters related to the general strategy because they resemble typical CNN optimization, such as Sauron, are more beneficial. As suggested by the reviewer, we will clarify this in the paper.

---

> > ### Comment · Reviewer_xHRW · 2022-08-07
> > **I understand, but...**
> >
> > I am satisfied with the reply to most of my comments and I understand the difficulty of illustrating the interpretability of feature maps in a condensed form, but it is not completely satisfying if this is the end of it.
> >
> > I do not understand the authors reply regarding hyperparameters and the relevance of separating these two types of hyperparameters as described.
> > - "it facilitates the implementation of our method and the tuning of Sauron's hyperparameters", how and why?
> > - "Thus, filter pruning methods that have no hyperparameters related to the general strategy because they resemble typical CNN optimization, such as Sauron, are more beneficial", for what? I don't think this is sufficiently clear.

---

> > > ### Author Response · Authors · 2022-08-08
> > > **Response**
> > >
> > > We thank the reviewer for reading and answering our response, and we are glad that the reviewer was satisfied with our reply to most of the comments. We hope to clarify our previous answer regarding Sauron’s hyperparameters.
> > >
> > > > * "it facilitates the implementation of our method and the tuning of Sauron's hyperparameters", how and why?
> > > > * "Thus, filter pruning methods that have no hyperparameters related to the general strategy because they resemble typical CNN optimization, such as Sauron, are more beneficial", for what? I don't think this is sufficiently clear.
> > >
> > > The lack of hyperparameters related to the design decisions of the general pruning strategy (e.g., "number of epochs before pruning starts", "pruning iterations") makes Sauron resemble typical CNN optimization (lines 96-98). Due to such resemblance, the implementation of Sauron is easy; in our implementation, Sauron is a plug-and-play module incorporated into nnUNet (Section 3.5). By contrast, consider other pruning approaches that have multiple pruning phases and require hyperparameters related to their specific optimization and pruning flow. Such approaches require much more effort to implement and deploy in existing applications because of their difference with respect to standard CNN optimization, whereas utilizing Sauron is as simple as copypasting our code provided in the supplementary material. In summary, having no hyperparameters related to the general pruning strategy, such as Sauron, is beneficial for implementing and deploying pruning strategies.

---

### Official Review · Reviewer_Vhi7 · 2022-07-10

**Rating:** 4
**Confidence:** 4
**Soundness:** 3 good
**Presentation:** 3 good
**Contribution:** 3 good

**Summary:**

The authors proposed a regularization term in the loss for the filter pruning in medical image segmentation tasks. The regularization term minimizes the difference of feature maps from each convolutional layer, which enforces the similarity of all features against the feature map of the first channel. The filter pruning is conducted by removing the filter producing similar features. Three medical segmentation datasets (with 3D volumes) are employed for the experiments, and Superior results of the proposed method are reported in comparison to previous feature pruning methods. I have several concerns about the presented work, mainly regarding the experimental results.

**Questions:**

It will be helpful to see the authors' response to my abovementioned concerns.

**Limitations:**

Good for the presented application.

**Strengths And Weaknesses:**

+ The proposed pruning method can achieve better results than prior arts, proving the add-on regularization term effectively promotes feature clustering.
+ The manuscript is overall easy to follow.

Weakness:
- The motivation for including the regularization term is not clear to me. For instance, I am not sure why the similarity of features is always enforced between the feature of the first channel and the rest. I did not see how the statement "... makes those feature maps near the feature map in the first channel O_1 even closer. At the same time, those features that are dissimilar to O_1 become more similar to other feature maps..." is achieved, especially how are the other clusters formed as in the second part?
- The proposed regularization term is rather straightforward, which is not technically innovative.
- Are the computed FLOPs computed in training or testing? It will be helpful to show the number of filters removed in the model and the exact number of parameters for the adjusted models.

---

> ### Author Response · Authors · 2022-08-02
> **Response**
>
> Thank you for your time and the positive assessment of our work.
>
> > * The motivation for including the regularization term is not clear to me. For instance, I am not sure why the similarity of features is always enforced between the feature of the first channel and the rest. I did not see how the statement "... makes those feature maps near the feature map in the first channel O_1 even closer. At the same time, those features that are dissimilar to O_1 become more similar to other feature maps..." is achieved, especially how are the other clusters formed as in the second part?
>
> The motivation for minimizing the proposed regularization term is to reduce the distance between feature maps in the clusters formed during CNNs optimization. We would like to emphasize that clusters of feature maps are formed regardless of any regularization term or pruning strategy (line 125). In consequence, such cluster formation phenomenon has motivated the development of filter pruning methods that eliminate redundant feature maps (see "Redundancy elimination" in the Previous Work section). Our experiments also corroborated the formation of clusters of feature maps in vanilla CNN optimization (Fig 1-b: feature maps are clustered after optimizing a CNN without the regularization term; Fig 1-e,f,g "Increase": some convolutional layers have higher clusterability after the optimization).
>
> In practice, it does not matter which feature map is chosen as a reference to minimize the distances in $\delta_{opt}$ (Eq. 2). We chose the first feature map as it was easy to implement; we will include this justification in the paper to improve its clarity.
>
> With the first feature map chosen as a reference (Eq. 2) and considering that the formation of clusters occurs regardless of any regularization term, minimizing $\delta_{opt}$ during the optimization will reduce (even more) the distance between the first feature map and all the other feature maps from the same cluster. Hence, we wrote "$\delta_{opt}$ regularization makes those feature maps near the feature map in the first channel O_1 even closer". Regarding the feature maps in other clusters, their within-cluster distance is also reduced because minimizing $\delta_{opt}$ equates to minimizing an upper bound of such distances. Hence, we wrote that "At the same time, those features that are dissimilar to O_1 become more similar to other feature maps from the same cluster".
>
> In summary, the formation of clusters of feature maps during the optimization of convolutional neural networks is inherent to CNNs optimization, and minimizing the proposed regularization term further promotes such cluster formation that we ultimately exploit for filter pruning.
>
> > * The proposed regularization term is rather straightforward, which is not technically innovative.
>
> We agree that the proposed regularization term is straightforward; this is similar to most influential regularization terms, such as L2. We think that complexity is not a necessary component of technical innovativeness, and given the outperformance and higher pruning rate achieved by our method, the straightforwardness of the proposed regularization term is a positive property that makes our method easy to follow, implement, and overall attractive.
>
> > * Are the computed FLOPs computed in training or testing? It will be helpful to show the number of filters removed in the model and the exact number of parameters for the adjusted models.
>
> We computed the FLOPs in testing. We thank the reviewer for the suggestion, and we will include the number of filters removed and the exact number of parameters after pruning.

---

> > ### Comment · Reviewer_Vhi7 · 2022-08-08
> > **Still not sure how the clusters are formed**
> >
> > I am still not convinced about "the proposed regularization term further promotes such cluster formation" by enforcing the similarity of features against the first one/channel, though the final results improved. Is there any way to analyze the clusters further? Maybe see what the clusters really represent?

---

> > > ### Author Response · Authors · 2022-08-08
> > > **Response**
> > >
> > > We thank the reviewer for the time and feedback provided.
> > >
> > > > I am still not convinced about "the proposed regularization term further promotes such cluster formation" by enforcing the similarity of features against the first one/channel, though the final results improved.
> > >
> > > As the reviewer indicated, we showed that Dice coefficient, HD95, and FLOPs reduction improved with the proposed $\delta_{opt}$ regularization (Section 4.1). Additionally, we showed that feature maps' clusterability also increased with $\delta_{opt}$ regularization (Section 4.2 and Figure 1). This is because minimizing $\delta_{opt}$ equates to minimizing an upper bound of the distances between the feature maps of the same cluster (lines 128-132). Let us illustrate this with an example.
> > >
> > > Consider a convolutional layer with five feature maps, $O_i : i \in $ {$1 , 2, 3, 4, 5 $}. These feature maps are grouped in two clusters: cluster A comprises feature maps $O_1, O_2, O_3$, and cluster B comprises feature maps $O_4, O_5$. By minimizing $\sum^{N}_{r=2} ||O_1 - O_r||_2$ (a simplified version of Eq. 2), where $N=5$, feature maps 1, 2 and 3 (cluster A) will become closer to each other (i.e., more similar) because the distance between feature maps $O_1$ and $O_2$, and feature maps $O_1$ and $O_3$ is reduced. Furthermore, because $||O_1 - O_4||_2 + ||O_1 - O_5||_2$ is an upper bound of $||O_4 - O_5||_2$ (lines 131-132), minimizing such upper bound reduces $||O_4 - O_5||_2$. In consequence, feature maps in cluster B also become closer to each other. We show this effect in Section 4.2 in different convolutional layers and with different $\lambda$ values.
> > >
> > > > Is there any way to analyze the clusters further?
> > >
> > > Cluster analysis, such as determining whether a dataset has clusters or finding the number of clusters, is challenging and it is a research topic on its own (Adolfsson et al. (2019)). In our work, we showed clusterability with three different metrics; one of them, the dip-test (Kalogeratos and Likas (2012)), was published in this venue.
> > >
> > > > Maybe see what the clusters really represent?
> > >
> > > We argue that each cluster can be thought of as a distinct operation and that the feature maps within each cluster can be viewed as noisy outputs of such operation (line 281). This view is supported by Figure 2 in Section 4.3, where we can see that, after pruning with Sauron, each feature map represents a different operation (e.g., the top-right feature map of Figure 2 corresponds to the operation "extract lesion from rat brain image"). In contrast, without pruning, we can see that there are several feature maps that perform the same operation (e.g., Appendix E, Figure S3, feature maps 2, 8, 16, 17 and 18 also correspond to the operation "extract lesion from rat brain image"). The goal of Sauron is to make those feature maps that correspond to the same operation closer to each other (i.e., more similar) by increasing feature maps clusterability via $\delta_{opt}$ regularization to, subsequently, eliminate all feature maps except one.
> > >
> > > We hope to have clarified Reviewer Vhi7's concern, and we are happy to discuss further if there remain any questions.
> > >
> > > Argyris Kalogeratos and Aristidis Likas. Dip-means: an incremental clustering method for estimating the number of clusters. Advances in neural information processing systems, 25:2393–2401, 2012.
> > >
> > > Andreas Adolfsson, Margareta Ackerman, and Naomi C Brownstein.  To cluster, or not to cluster: An analysis of clusterability methods.Pattern Recognition, 88:13–26, 2019.

---

### Official Review · Reviewer_qmVv · 2022-07-10

**Rating:** 5
**Confidence:** 4
**Soundness:** 3 good
**Presentation:** 3 good
**Contribution:** 2 fair

**Summary:**

This paper tackles filter pruning on deep convolutional networks for medical image segmentation. Training loss is augmented with a regularization term that penalizes distances between feature maps and forms clusters, and thus, pruning does not introduce extra training parameters or phases such as pre-training. Feature maps whose distances are below a threshold are then dropped, where the thresholds are adaptive based on layer and epoch. This strategy is used for training the state-of-the-art nn-UNet on 3 benchmark medical image segmentation datasets. The resulting pruned models maintain the segmentation performance of unpruned models and outperform other pruning methods in terms of segmentation performance and inference time.


**Questions:**

N/A

**Limitations:**

Limitations and potential impacts are discussed.

**Strengths And Weaknesses:**

Strengths: Pruning deep neural networks for medical image segmentation is an important problem due to high dimensionality and long training times. The paper is well-written, method and experiment setup are clearly explained.

Weaknesses: Following recent and similar works are not mentioned in related works/experiments, hindering the clarity of novelty. Particularly, Zhou et al. (2020), Chen et al. (2020) and Dinsdalea et al. (2022) focus on filter pruning for medical image segmentation; it is suggested that the authors compare/contrast their approach with these works.

Zhou et al. “Evolutionary Compression of Deep Neural Networks for Biomedical Image Segmentation” (2020)

Chen et al. “α-UNet++: A Data-Driven Neural Network Architecture for Medical Image Segmentation” (2020)

Dinsdale et al. “STAMP: Simultaneous Training and Model Pruning for Low Data Regimes in Medical Image Segmentation” (2022)

He et al. “CAP: Context-Aware Pruning for Semantic Segmentation” (2021)

Ditschuneit et al. “AUTO-COMPRESSING SUBSET PRUNING FOR SEMANTIC IMAGE SEGMENTATION” (2022)

Chen et al. “MTP: MULTI-TASK PRUNING FOR EFFICIENT SEMANTIC SEGMENTATION NETWORKS” (2022)

Sabih et al. “DyFiP: Explainable AI-based Dynamic Filter Pruning of Convolutional Neural Networks” (2022)

---

> ### Author Response · Authors · 2022-08-02
> **Response**
>
> Thank you very much for your positive feedback and useful suggestion. We agree, and we will add to our Previous work section those filter pruning methods applied to medical/biomedical image segmentation tasks since they share the same focus as our paper.

---

### Meta-Review · Area_Chair_LDGM · 2022-08-26

**Recommendation:** Reject
**Confidence:** Certain

**Metareview:**

The paper proposed a method for pruning filters in image segmentation networks by removing filters during training that are closely clustered. Unlike prior works, the approach is described as single-phase, meaning it prunes during normal training. To obtain smaller networks, a term which promotes feature map clustering is added to the loss. The experiments uses nnUNet as a baseline network and 3 other recent network pruning methods as comparison. The results show good performance (Dice/HD95) with more pruned networks on three medical segmentation datasets (with 3D volumes), and largely reduced FLOPs.

Deep neural networks pruning for medical image segmentation is an important problem due to high dimensionality and long training times.  The paper is well-written, method and experiment setup are clearly explained.

The contribution was nevertheless found to be limited among a very fast growing literature.
The core contribution could be better explained.
The authors mostly proposed a general filter pruning without any specific optimization for U-Net series. It is thus necessary to compare Sauron against more existing filter pruning methods.

The contribution is fair, but that it remains overall a bit limited for NeurIPS, in the sense that it does not offer strong insights. Besides, the effectiveness of Sauron is also questionable on 3D U-Net, where the pruned model fails to provide satisfactory performance.


**Award:**

No

---

### Decision · Program_Chairs · 2022-09-14

Reject